# Post-Soviet Suburbanization as Part of Broader Metropolitan Change: A Comparative Analysis of Saint Petersburg and Riga

Guido Sechi [1,*][iD], Dmitrii Zhitin [2][iD], Zaiga Krisjane [1] and Maris Berzins [1,*][iD]

1 Department of Geography, University of Latvia, LV1004 Riga, Latvia; zaiga.krisjane@lu.lv
2 Department of Economic and Social Geography, Institute of Earth Sciences, St. Petersburg State University, 199034 St. Petersburg, Russia; d.zhitin@spbu.ru
* Correspondence: guidosechi78@gmail.com (G.S.); maris.berzins@lu.lv (M.B.)

**Abstract:** Studies on post-socialist suburbanization, which originally focused on demand side dynamics and linear narratives of modernization, have progressively adopted more holistic approaches that consider the various dimensions and factors behind the phenomenon. However, there are still significant gaps and shortcomings affecting this research domain; studies encompassing demand side and supply side dynamics are rare, and so are comparative perspectives. The phenomenon has rarely been analyzed in the context of broader metropolitan change, together with other dynamics such as inner-city gentrification, degradation, or maintenance/regeneration of socialist era residential neighborhoods. This study addresses the mentioned gaps through a multi-dimensional comparative pilot analysis of suburban dynamics in Saint Petersburg and Riga. The analysis encompasses the spatial extent of demographic, socioeconomic, and housing market dynamics within the broader context of metropolitan change. The findings reveal a picture of a demographically and economically significant phenomenon with remarkable implications for macro- and micro-level socio-spatial segmentation; the distinctive features between the two cases are primarily due to migration dynamics and the short/medium term effects of the 2008 financial crisis on the real estate market and industry. Overall, the suburban option appears to be an attractive option for the demand side (in terms of an economic trade-off or societal aspiration) as well as a safe and profitable option for developers.

**Keywords:** suburbanization; urban social change; comparative urbanism; real estate market; metropolitan area

## 1. Introduction

Studies of socio-spatial change associated with transformations after 1989/1991, in the cities of Central-Eastern Europe (CEE) and the former Soviet Union (FSU), have provided many relevant contributions to the description and identification of the plurality of ways in which socio-spatial segmentation phenomena as diverse as segregation, gentrification, sub- and de-urbanization develop in the post-socialist city, for example, [1,2]. However, most studies in this domain have generally been empirical and have focused on specific territorial contexts. This has contributed to fostering path dependency-based explanations and relatively linear narratives of modernization. Recently, this dominant approach has been criticized by some scholars for overlooking multi-scalar dynamics and comparative approaches, which are both seen as beneficial to moving from purely descriptive and inductive contributions to critical theoretical-conceptual contributions [3]. The consensus in related regional and social science domains corroborates such criticisms. Since the 1990s, post-socialist anthropology has significantly contributed to a critical understanding of socialist characteristics and transition dynamics by shedding light on ideological, spatial, and economic dynamics, which has ignited a lively debate about the benefits and rationale of comparative vs. context-specific analysis, for example, [4,5]. Moreover, empirical evidence of increasing spatial inequalities and polarization in Europe, and particularly in post-socialist countries, has led to a recovery of multi-scalar analyses of spatial development

proposed by critical geographers and spatial thinkers versus purely 'territorial' perspectives, for example [6].

Urban space and its transformations play a central role in the study of post-socialist transition, considering the specific connections which existed under state socialism among the built environment and ideology, society, and welfare [7]. The emblematic role that the transformations of the built environment assume not only concern societal shifts, but also ideological, economic, and institutional change [8,9].

One of the most studied phenomena in post-socialist socio-spatial transformations is suburbanization dynamics, defined as the reshaping of an urban region 'when suburban growth outpaces the growth of the core city' [10]. Residential suburbanization in the 'Western' understanding of the term was a negligible phenomenon under state socialism, but it has become very relevant in the CEE and FSU region since the collapse of the socialist system. Until a few years ago, post-socialist suburbanization was generally regarded as a modernization phenomenon, albeit associated with the significant detrimental effects of urban sprawl [10]. Indeed, the trend towards suburbanization in the region since the early 1990s represents a significant shift from relatively compact residential patterns, based on large housing estates for almost four decades since the late 1950s, towards decentralization; as such, it raises significant sustainability issues in terms of land consumption [11] and other environmental impacts of suburban settlements [12].

From the 2000s onwards, scholars have started paying attention to the socioeconomic composition of the suburban migrant population, highlighting the relevant implications of the phenomenon in terms of socio-spatial differentiation and polarization [13,14]. Since the 1990s, a prevalent consumption-oriented approach focused on societal change has coexisted with a less common production-oriented approach. Albeit both approaches have considered the co-existence of local specificities and global market capitalist trends, the latter approach has more actively emphasized the relevance of the globalized capital logic in shaping suburban landscapes and socio-spatial dynamics [15]. A perspective framing socio-urban changes within broader dynamics has emerged in the 2010s, advocating the necessity to interpret societal and urban change within institutional and ideological transformations associated with transition [8,16]. It is now widely accepted that suburbanization in the post-socialist region is a complex phenomenon rather than a simple matter of individual choices, encompassing dimensions of ideological, economic, and infrastructural space transformation [10,17].

Still, it can be argued that post-socialist suburbanization studies (and, to a lesser extent, post-socialist urban change studies in general) are hampered by a lack of systematic analyses that are capable of encompassing the consumption side and the production side, i.e., the social dimension of the phenomenon on the one hand, and the spatial and economic changes on the other hand, and their interrelationships (1), and framing specific instances within broader processes of urban change (2).

The global debate on suburbanization is intrinsically connected to the topic of neoliberal planning and ideology. For instance, the U.S. suburban trend from the 1970s onwards has been described as an explicitly anti-urban phenomenon [18] with a strong conservative grassroots component [19] and has been regarded to be one of the main spatial embodiments of the embracement of deregulation and rejection of Keynesianism [20]. A complementary and related phenomenon to urban suburbanization, i.e., gentrification, offers useful parallels and analytical tools. The more than 30-year-old global debate that has included the confrontation of production side-oriented and societal change-focused explanations of this phenomenon [21–24] has led to the common acknowledgement that urban change is a complex process which encompasses economic, spatial, socio-cultural and political-institutional dimensions. In addition, this debate has also evidenced that the crucial strength of critical perspectives focused on capital investment and profitability from the production side lies in their capability of framing specific dynamics within more extensive processes [25,26]; both these tenets inform our study that investigates, in

a multi-dimensional way, suburbanization trends and dynamics in two large post-Soviet cities, Saint Petersburg and Riga.

We conceptualize suburbanization as a phenomenon that needs to be framed and analyzed in broader urban dynamics and a process encompassing a dialectic between production side strategies and societal change. In other words, on the one hand, we hypothesize that the aspects on which we focus are relevant manifestations, at the urban level, of the social transformations associated with the impact of market capitalism on the inherited socialist and pre-socialist characteristics of urban and suburban space [27]. On the other hand, we acknowledge that socio-spatial transformations are complex phenomena that encompass spatial, economic, and social change dynamics. Thus, this point is addressed by discussing the data and results in the context of the broader urban development trends in the two cities under exam.

A comparative analysis of post-socialist cities can be viewed as a way to incorporate similarities and specificities. Some authors point out how recent post-socialist urban studies have mostly focused on the peculiarities of national contexts [26]. These attitudes have been motivated mainly by a correct rejection of Orientalist and Cold War frameworks which tended to overgeneralize and overlook such peculiarities, but have also led to conceptually overlooking the specificities of the socialist city and the socio-spatial system vs. the Western capitalist one [28]; this may consequently lead to underestimating the ideological and political dimension of post-1991 urban transformations.

Urban scholars regard a comparative analysis as beneficial for stimulating critical thinking [29,30]. Comparative analyses of urban contexts characterized by the common heritage of socialism can help to acknowledge the role of ideology and ideological shifts, with all the relevant consequences in terms of politics of space, and, in this way, to provide conceptual insights [3] while at the same time acknowledging contextual peculiarities and trajectories during and after the state socialist experience. Moreover, the comparative perspective may help to address some shortcomings of the literature on post-socialist urban change, where specific phenomena are rarely framed within the overall logic of neoliberal urban governance, the historical evolution of neoliberalism, and its global tendencies [31].

However, an exclusive focus on the post-socialist context may result in an entrenchment focused on specific peculiarities. Hence, in our study, we frame our comparative analysis within the broader debates about urban change, social segmentation, and uneven development.

In particular, we attempt to shed light upon the following research questions:

- What are the suburbanization patterns in the two metropolitan areas in terms of demographic, socioeconomic, and housing market trends over the most recent decade/-s?
- What are the main drivers of suburbanization in the analyzed metropolitan areas?
- How can the overall dynamics of suburban development be interpreted based on the analyzed dimensions and in the light of broader urban change processes in the two cities? What are the post-socialist urban change theory implications based on observed trends and dynamics?

Due to limitations in the availability of comparable data, our study must be considered to be a pilot empirical analysis aimed at defining possible directions to address conceptual and methodological gaps in the literature.

This paper is articulated into seven sections. In the next section, we outline the literature debate on post-socialist suburbanization, framing it within the broader issues of urban change in transition, and outlines the rationale for comparative urbanism in post-socialist studies. In Section 3, we outline the study context by describing the characteristics of spatial and urban planning in the Russian Federation and Latvia and the patterns of metropolitan change in Saint Petersburg and Riga. In Section 4, we describe the materials and methods including the spatial extent of the case study areas. In Section 5, we summarize the results of the study and, in Section 6, we discuss the results from a comparative perspective and within the broader context of urban change. Finally, in Section 7, we outline the conceptual implications of our analysis and possible guidelines for further research.

## 2. The Suburbanization Debate in the Context of Post-Socialist Urban Change

The post-socialist suburbanization debate initially counterposed scholars who regarded the phenomenon as basically akin to the trends common to Western capitalist countries and those who emphasized some peculiar characteristics rooted in path dependency [32] and different economic logics [33]. Later on, the debate developed along two different axes: the emphasis on societal vs. production side drivers and the transitional vs. hybrid conceptual view of post-socialist urban change.

Studies that have been carried out in the last two decades have found evidence of the significant implications of the phenomenon in terms of socio-spatial differentiation and polarization. For example, suburbanization has been found to be an upper-class phenomenon in Budapest [34] and in Prague [35]. Sharp micro-level polarization due to the affluence of wealthy city dwellers in traditionally peri-urban poor settlements has been witnessed in Sofia [36]. Studies about theTallinn metropolitan area have described a more complex picture about the social profile of suburbanites, but the trend has been, nevertheless, found to have substantially increased polarization in terms of socio-economic status while at the same time reducing segregation in terms of educational status [13,14]. In addition, a few studies have adopted a multi-scalar approach focused on the role of capital in shaping post-socialist spaces instead of a consumption-based approach. For instance, a study of 1990s suburbanization in Hungary [15] found evidence of aggressive interest representation of developers and non-socially oriented housing policies as main drivers behind the development of highly polarized suburbs. Market-friendly policies on land use have also been called into question with regard to the issue of sustainable land consumption [37]. Some studies seem to point out that, notwithstanding some contextual specificities, the logic at work in post-socialist cities is not significantly different from that operating in 'Western' cities and the dynamics under examination are basically explained as the impact of neoliberal destruction and creation mechanisms on territory and space [38]. In addition, some authors who share this multi-scalar perspective argue that suburbanization in some post-socialist contexts is influenced by a significantly different political economy of place than in 'Western' countries, emphasizing hybridization over fully globalized processes, for example, [39]. Regardless of these differences, multi-scalar approaches have pointed out other detrimental effects of post-socialist suburbanization in addition to sprawl, such as social tensions, segregation, and exclusion. Suburbanization and gentrification, in general, are considered to be relevant processes of post-socialist urban socio-spatial change that affect socioeconomic segregation [40].

There has been a certain degree of convergence between societal and production side perspectives in that it is, by now, widely accepted that suburbanization in the post-socialist region is a societal phenomenon rather than a matter of individual choices and cannot be interpreted without acknowledging the ideological influence of the post-1989/1991 neoliberal paradigm [8,17]. However, this also implies that in order to provide a more complete picture of the phenomenon, societal change must be examined within the context of, and alongside, the dynamics of economic change and physical transformation [16,41]. In this regard, the literature agrees on the hybrid characteristics of post-socialist physical environments; in the suburban context, spatial-physical change is associated with the co-existence of persisting socialist structures, elements of transition and transformation, and new suburban and post-suburban spatial (infra)structures [32]. However, hybridization can be explained in different ways. There is a conceptual difference between views according to which spatial/infrastructural change is a relatively slow but inevitable and basically linear process that temporally follows institutional, and then societal transformations [8], and views that postulate an inherent hybrid nature of post-socialist transition, characterized by capitalist subsumption of socialist infrastructures [27].

The post-socialist suburbanization debate can also be framed within the global discussion on urban change determinants; in particular, to a certain extent, it echoes the decades-long scientific controversy about the causes of gentrification in the North American continent [21,22], which expressed conflicting views between a Marxist/critical theory

focused on economic determinants (capital, land, and housing market) and a 'postmodernist' view focused on societal change. By the mid-1990s, the complementarity of these two perspectives was widely acknowledged [42] and a substantial consensus conceptualizing gentrification as a multi-faceted process of class transformation emerged; however, controversies about the role of capital investment vs. societal 'professionalization' quickly re-appeared, and are still relevant to this day [24]. However, these controversies are probably better explained as the tendency to describe a specific urban change phenomenon either as an isolated event or within the context of broader urban, metropolitan, and global processes [43].

It is important to consider the views of transition within which contributions to the study of post-socialist urban change are conceptually framed to understand and evaluate these phenomena. In the field of urban studies, recent conceptual contributions [44,45] identify three main views of 'post-socialism':

-       As a transitory stage from socialism to capitalism, which implies an inevitable convergence towards a Western model, albeit delayed or altered by socialist legacies;
-       As a hybrid phenomenon characterized by the inter-mingling of capitalist and socialist elements, resulting in different trajectories and forms of urban change;
-       As a phenomenon to be understood in a 'de-territorialized' perspective, emphasizing post-socialist discontinuities next to socialist legacies, favoring comparative analysis from a global perspective and defining post-socialist cities as complex entities.

It can be argued that most of the suburbanization literature generally adheres to the 'transitory stage' framework, albeit recognizing the existence of hybrid characteristics and trajectories. Hence, however nuanced, this approach is nonetheless influenced by a relatively linear view of the transition to capitalism as an inevitable modernization/Westernization process, albeit often complex and slowed down by 'legacies' resisting eradication.

Recent conceptual contributions outline a fourth perspective on the post-socialist city [3,9,27,31], which present points in common with the hybridization and the de-territorialized view and advocate a conceptual and critical analysis of the neoliberal transition ideology, implying the need for multi-scalar, multi-dimensional, and comparative approaches. Some of the proponents of this view have emphasized—in a conceptualization indebted to critical spatial theorists of the late 20th century, for example, [46,47]—that the persistency of spatial legacies should not shed the dominance of the neoliberal logic and ideology in shaping spatial, both formal and functional, change; and that the emphasis on the different pace of institutional, societal, and spatial transformations risks obscuring the complex dialectics existing among these levels. In this perspective, socio-spatial segmentation phenomena, similar to suburbanization and gentrification, translate economic inequalities into social status, influenced by socioeconomic change and housing market strategies [9]. Other authors emphasize the necessity of addressing post-socialist urban transformation in a multi-dimensional and holistic way, moving beyond the focus on narrow aspects of neoliberal urban policies, and taking into account their dynamic and transformative nature from 1989/1991 onwards [31]. In this logic, a study of post-socialist suburbanization would require a multi-scalar and multi-dimensional approach, encompassing the connections between the overall logic of neoliberal development and the various forms of urban transformation and the social, economic, and political-institutional dimensions that such changes encompass.

## 3. The Study Context

### 3.1. Spatial Planning and Development in the Russian Federation and Latvia

The Soviet spatial planning system operated in the context of a development strategy and logic mostly privileging urban and industrial development. As such, phenomena comparable with suburbanization in USSR times were mainly associated with in-migrants from rural and urban areas searching for jobs in the industrial sector. However, in some areas, such as the Baltic countries, where collective farming was generally success-

ful and supported by good residential infrastructure, outmigration from cities was also relatively significant [11,48].

The collapse of the USSR meant the immediate transition from a rigid and redistributive centralized system, lacking spatial planning legal rules, to a capitalist and decentralized system. The strategic priorities of economic restructuring, the slow adaptation of the legislation to the new context, and the extreme ideological backlash against state intervention led to a profound crisis of spatial planning for a few years, with partial signs of recovery only after 1995 [49]. This led to the emergence of unregulated suburbanization and sprawl dynamics during the 1990s [50], with residents of a lower socioeconomic status moving to suburban areas searching for cheaper housing [14]. Upper- and middle-class suburbanization is generally a post-2000 phenomenon, favored, in practical, by mortgage availability, increase in wealth, and supply of new housing, which had been basically interrupted in the previous decade. Suburbanizing trends have indeed been growing in the 2000s decade in the hinterland of the two cities which are the objects of this study, Saint Petersburg and Riga [14,51].

Looking at the last 20 years, Latvia and the Russian Federation have shown differences and similarities when analyzed from the perspective of societal change. Whereas in Latvia, as in most CEE countries, the population increase in the Riga hinterland is mainly associated with city residents [14], in the largest Russian cities it is also significantly boosted by internal migration from more peripheral areas and external migration from other countries of the Commonwealth of Independent States (CIS), Central Asian countries in particular [52], making it a largely heterogeneous and socially polarized phenomenon. However, suburbanization in Latvia is also rather polarized in terms of socioeconomic composition, with a prevalence of high- and low-status rather than middle-class suburbanites [14].

Analyses of the impact of neoliberal development strategies and mechanisms on suburbanization dynamics in the two countries are generally rare. Some studies have emphasized the hybrid characteristics of the phenomenon in Russia, i.e., the co-existence of 'improvised' and global economic mechanisms and strategies affecting suburban development, with the latter gradually replacing the former [53]. For example, the suburbanization phenomenon in Moscow is mainly driven by private investors' profit collections [52]. An analysis of the case of Khimki, a Moscow region town de facto located within Moscow city area, points out the relevant influence of the relations between local and federal subject authorities in shaping development paths, and the prevalence of speculative practices and sectoral planning over long-term planning and consistent place-making strategies [39]. In Latvia, studies of suburban development in the Riga metropolitan area during the 2000s decades have highlighted the prevalence of market-oriented and speculative logic, as well as insufficient involvement of local authorities; one relevant consequence has been the scattered nature of the settlements that have been officially recognized, ex-post, in spatial plans as villages, but lack common infrastructure and functional consistency [54].

### 3.2. Metropolitan Change in Saint Petersburg and Riga

The logic inherent in our approach regards suburbanization as part of larger transformation processes within the metropolitan areas of both cities over the past three decades. Demographic dynamics in the two cities and metropolitan areas need be framed within the context of city decline as a common trend in Central-Eastern Europe and the western part of the former USSR since the early 1990s [55], but it must be considered that forms and characteristics of such decline are significantly heterogenous [56]. Sharp center-periphery dynamics due to strongly uneven spatial development, in many cases, including the Russian Federation and Latvia, have created a centripetal effect, with a polarization between growing or depopulating major cities with growing metropolitan areas and shrinking de-industrializing peripheral cities. Whereas St. Petersburg has been growing in the last years due to significant in-migration, the decline in Riga city is due to natural balance rather than migration rates, although the latter have been, to an extent, affected by the peripheral position of the country, in its turn, in the European Union.

The morphological structure of the two cities is broadly typical of historical large urban centers in Central-Eastern Europe and the western part of the former USSR, with a 'historical', pre-Soviet core surrounded by large post-Stalinist Soviet microdistricts consisting of panel housing estates, in their turn, surrounded by a heterogeneous suburban area. The differences lie in the significant developments that affected Saint Petersburg in early Soviet times that are spatially reflected in a layer between the historic core and the mass housing microdistricts, consisting of constructivist estates built around industrial plants and large Stalinist avenues, and in the much more urbanized character of its suburban area. As of the early 2010s, high social status residents in both cities were mainly concentrated in parts of the pre-Soviet historical core and historical prestige villa districts in the outer city (plus Stalinist era avenues in the case of Saint Petersburg) [57,58]. Saint Petersburg and Riga have both witnessed fragmented gentrification and micro-level polarization dynamics in their historical cores [57,59]. In Saint Petersburg, the fragmented nature of gentrification is associated with the high fragmentation of property, associated with the privatization program of the early 1990s, and strict heritage preservation rules that make large-scale renovations risky for developers [45]. In Riga, where the 'fragmentation' factor has been, to an extent, a lesser constraint, the fast city center development trends of the 2000s decade were halted by the 2008–2009 private debt crisis and the following sharp contraction of the construction sector and market [60], with a partial recovery only in the last few years.

In both cities, structural issues of physical degradation and lack of maintenance of post-Stalinist Soviet large housing estates can be observed [61,62]. In general, renovation of these estates is regarded to be unprofitable by real estate developers. In Riga, in-fill development in greener and more attractive Soviet neighborhoods is more common than renovation [63,64]. In Saint Petersburg, a recent plan for large scale renovation of Khrushchev-era housing estates has failed due to the fragmented ownership structure, resulting in a significant veto power of opposing residents and pointing to the existence of significant risks for profit-oriented developers in these areas [65]. In Riga, where EU funding is theoretically available for mass housing renovation, there have been no studies concerning constraints and profitability from the point of view of developers, but property fragmentation and shared land ownership are widely mentioned as obstacles to structural renovation [61].

Overall, the picture that emerges from the 'compact city' development trends in both cases is one where some significant constraints to profit-oriented large-scale renovations do exist, and construction of new housing, when feasible, appears to be a safer enterprise for private developers. The high-profit-/low-risk-driven preference for in-fill and especially suburban development at the expense of existing compact city residential infrastructure significantly increases land consumption within and outside city limits, in explicit or implicit contradiction with official sustainable development goals [66,67]. In Riga, where significant investment in city center infrastructure has started in recent years, 'rejuvenation' of the inner city has been observed, suggesting a reprise of gentrification and consequent re-urbanization [64]. The analysis of suburbanization trends needs to be interpreted within this broader context.

## 4. Materials and Methods

### 4.1. Methodology

The methodology adopted in this study descends from a view of urban changes as complex processes which encompass economic, spatial-infrastructural, socio-cultural, and political-institutional dimensions [25] and on the ambition of moving beyond purely descriptive accounts of post-socialist socio-spatial transformations by looking at causes and interrelated dynamics [3]. The core analysis of this study encompasses the production side and the consumption side, adopting a rationale first outlined following the debate on gentrification dynamics in North American cities in the late 1980s [21,22], but extended further to other domains of urban change, including post-socialist suburbanization [15,16,52]. This

approach implies an analysis of spatial economic and social aspects of urban change and their interrelations [21,23,25].

Our analysis covers the past ten to twenty years, depending on data availability and comparability. Since the 1990s meant a rather stiff housing market investment landscape in the context of harsh economic restructuring, the 2000s saw an increase in developer investment and renovation of the housing supply, accompanied by evidence of increased socio-spatial differentiation [57]. The 2010–2019 decade started with signs of gradual recovery from the 2008–2009 financial crisis, which had dramatic socioeconomic consequences and hit the construction and real estate sectors the most. Contraction figures have been more pronounced in Latvia, with a decrease in GDP of almost 18% [68,69] and a corresponding reduction in the house price index by more than 29% in 2009. In the context of a GDP contraction, the Russian economy and the construction sector have also sharply suffered, with a substantial decrease in mortgage crediting, price index, capital investment, and cost level growth [70,71]. However, the housing market's recovery has been faster in Russia due to a securitized finance system of state-owned banks that reduced exposure to financial risk [72]. In one of the few post-crisis studies on suburbanization in Russia, a shift from elite to middle-class suburbanization in large cities in the first years after the crisis was evident [73]. In Latvia, a post-crisis shift to more prudential international norms and lending has unintendedly led to a 'retreat' from housing financialization [74].

The housing market and socio-spatial trends and dynamics in the two countries following the crisis have not been systematically studied yet. Their analysis could be of interest, in terms of the post-socialist context and the global context, in order to grasp production side strategies as well as consumption side trends following a macro- and micro-economic shock and the re-configuration of the residential real estate sector and market.

The current study relies mainly on existing quantitative secondary data at the municipal or submunicipal level by focusing on suburbs of both metropolitan areas. Thus, we address various dimensions of (sub)urban change in Saint Petersburg and Riga. The comparative and multi-dimensional approach conceptually frames the research within a multi-level and, to an extent, multi-scalar perspective; in this regard, it represents an attempt to move beyond context-specific, purely descriptive approaches of socio-spatial change to provide material for theoretical-conceptual contributions, as advocated in the last years by several post-socialist urban scholars, for example, [3]. The following dimensions, measured at the municipal/submunicipal level, have been considered:

- Demographic trends, to identify their internal differentiation;
- Variation in average socioeconomic wealth, to grasp social segmentation dynamics;
- Variation of the housing supply and real estate market trends, to grasp production side-driven spatial and economic dynamics.

Russian and Latvian sociodemographic and housing-related data present significant differences in availability and aggregation levels. As much as possible we used comparable and updated data.

*4.2. Spatial Extent of the Selected Metropolitan Areas*

We have focused on suburbs of both cities or areas outside the administrative boundaries of Saint Petersburg's federal city and Riga's city but functionally connected to the core cities. In both cases, the metropolitan areas consist of functional urban areas determined by the extent of the urban daily commuting, which has been extensively studied in the case of Riga [75,76], but no relevant data are available on urban daily commuting in the Leningrad Oblast. At the same time, the Saint Petersburg metropolitan area is functionally diverse, with a different system of administrative divisions than the Riga metropolitan area.

The Saint Petersburg suburban area includes parts of two federal subjects (top-level political divisions of Russia): Saint Petersburg federal city and the Leningrad Oblast (region). Federal subjects are divided into districts that are divided into municipalities. The intra-urban division of Saint Petersburg federal city includes urban neighborhoods (municipal okrugs), municipal towns, and municipal settlements (villages). In Leningrad

Oblast, municipalities include towns, urban settlements (consisting of an urban center and rural areas), and rural settlements. The federal city of Saint Petersburg incorporates, next to the city proper, other towns and settlements that represent a relevant part of the suburban area.

The Saint Petersburg metropolitan area consists of the core city of Saint Petersburg and 32 municipalities within the federal city, and 79 municipalities located in the Leningrad Oblast. For this study, four functionally distinct settlement types were distinguished in the suburbs of the Saint Petersburg metropolitan area: (1) old industrial centers with negligible new development; (2) second home areas and recreational settlements, including pre-1991 resort settlements; (3) areas of new mass housing development; (4) rural areas with negligible development. The first group includes 23 municipalities of Soviet-era urban areas dominated by industrial or transport enterprises (Figure 1).

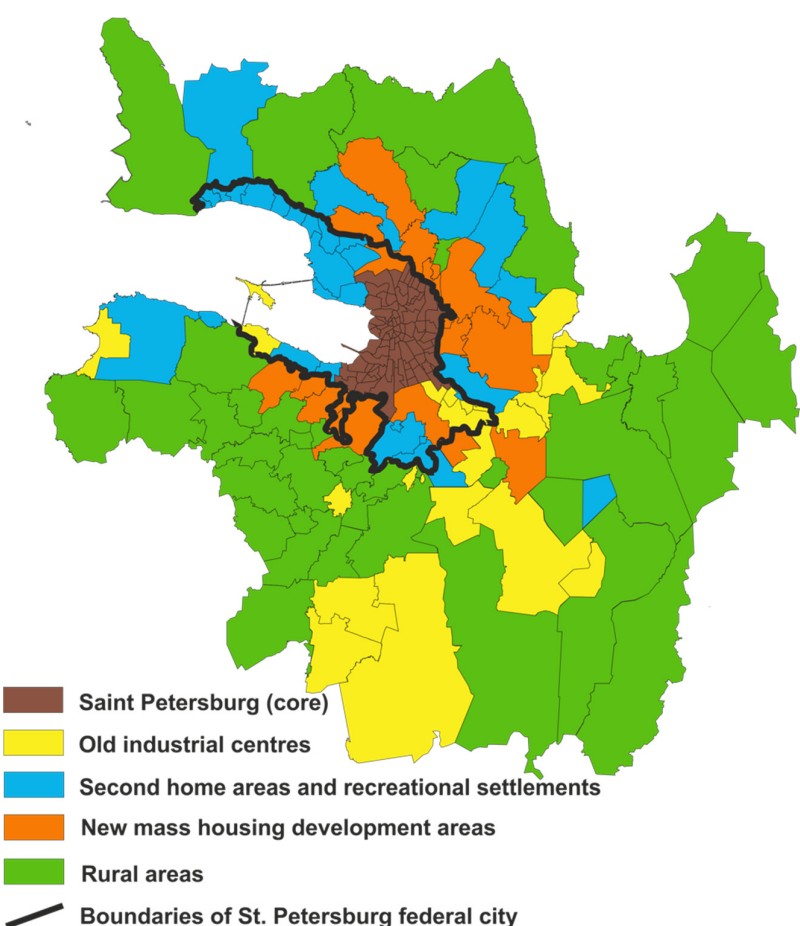

**Figure 1.** Saint Petersburg metropolitan area and its functional division.

In the post-Soviet period, most of these enterprises experienced a severe economic decline. The second group of 31 urban or rural municipalities includes traditional resort towns and settlements that successfully adapted to post-Soviet changes. Following the economic growth of the early 2000s, these municipalities began to be actively developed by the emerging middle-class residents of Saint Petersburg. Single housing development in the last decades also targeted traditional rural areas. The rapid growth in housing construction over the past two decades has led to the formation of new mass housing development areas (18 municipalities). This development has generally targeted traditional rural areas close to the core city. The fourth type of municipalities includes rural areas with limited or negligible new development on the periphery of the metropolitan area. These municipalities are the least affected by post-Soviet urban transformation and remain

as potential 'reserve territory' for residential development. It is worth noting that some municipalities fall somewhere in the middle of the specified types in terms of functionality.

The Riga metropolitan area is the most densely populated urban area in Latvia and has experienced significant population growth and concentration in recent decades.

In Latvia, the main administrative unit is the municipality. This category encompasses major cities and other municipalities, which either include urban and rural areas, or only rural areas. The Riga metropolitan area includes 11 municipalities and 31 submunicipal or territorial units, consisting of 10 urban areas and 21 rural territories. Apart from Riga, the metropolitan area includes another major city, the historical resort town of Jurmala. The other suburban towns in the metropolitan area are considerably smaller and have mainly developed as satellite towns of Riga. Despite the diverse functional origins of these towns from a historical perspective, we distinguished them all together with Jurmala as a distinct settlement type in the following analysis (Figure 2). Thus, our study, in the case of suburbs of the Riga metropolitan area, comprised only two different settlement types: (1) traditional urban centers with negligible or limited new development and (2) rural areas with significant suburban residential development. The first group includes urban and semi-urban areas where industry and services have already been established since the Soviet period, but most of the population has been employed in the non-agricultural sector. The second group consists of traditional rural settlements that more easily adapted to post-Soviet conditions.

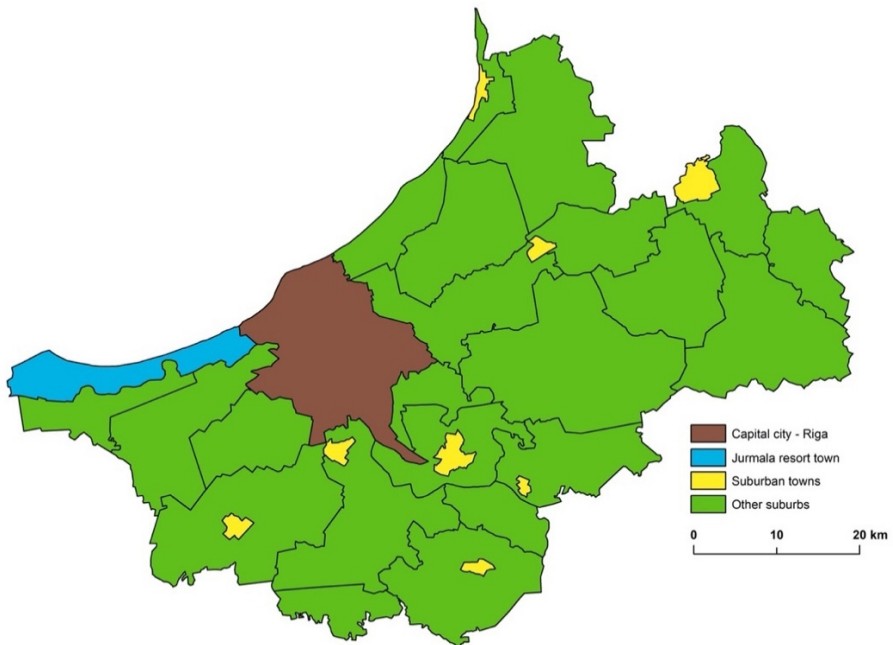

**Figure 2.** Riga metropolitan area and its functional division.

## 5. Results

The metropolitan areas discussed in this study are both very different in size, measured in terms of population. The Saint Petersburg metropolitan area, with nearly 6.5 million inhabitants in 2020, is more than seven times the size of the Riga metropolitan area, with just under 870,000 inhabitants. At the same time, it should be noted that in both metropolises, about 70% of the population of the entire metropolitan areas lives in the core cities, which shows a significant population concentration and mono-functional urban structure. Despite this, both metropolitan areas have been characterized by ongoing suburbanization processes over the past two decades, with population growth in the suburbs (Figure 3).

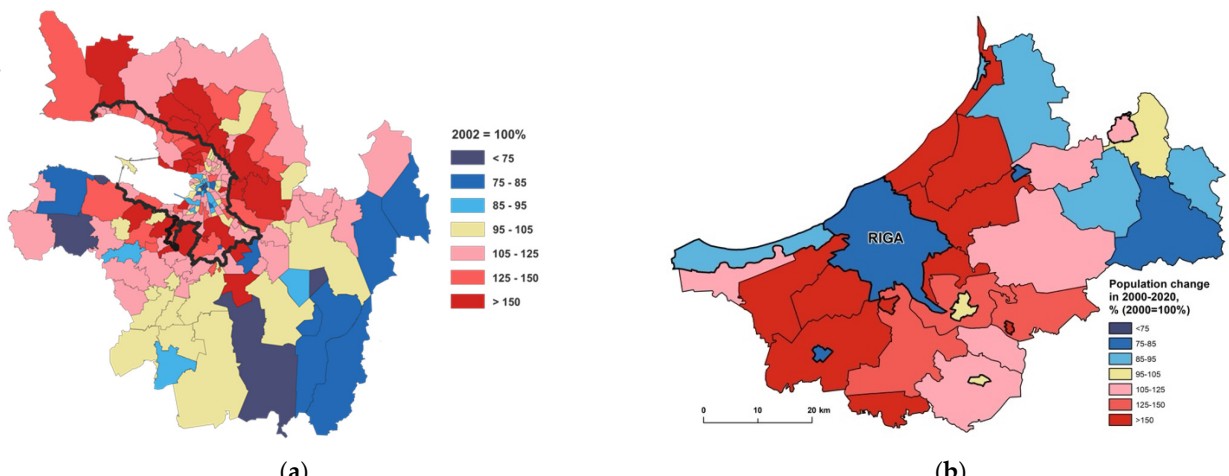

**Figure 3.** Population changes in: (**a**) Saint Petersburg metropolitan area, 2002–2020; (**b**) Riga metropolitan area, 2000–2020.

Over the past two decades, the population in the suburbs of the Saint Petersburg metropolitan area has grown at a much more rapid pace (41.6% increase) than in the urban core (11%); 70% of suburban municipalities have substantially increased their population, while population decrease has been observed only in municipalities located on the periphery of the metropolitan area. In contrast, the Riga metropolitan area shows marked differences between the core city and the suburban area. In the period under consideration since 2000, Riga has lost almost a fifth of its population (19% decrease), while the suburbs have, on average, experienced the same level of population growth.

In the following sections of the results, we look separately at both metropolitan areas, describing in more detail the differences in demographic, socioeconomic, housing supply, and market trends that correspond to the functional and geographical divisions we have adopted in our analysis.

### 5.1. Demographic Trends

**Saint Petersburg metropolitan area.** The population dynamics in the Saint Petersburg metropolitan area between 2002 and 2020 show some significant differences. Relatively less population growth was observed in the old industrial centers, increasing by only 6.6%, and in rural areas by 10.5%. In contrast, the increase in dacha-recreational municipalities was 27.1%, but the most remarkable growth, i.e., more than 2.6 times, was observed in the new high-rise areas. In the urban core, the number of residents increased by 11.0%. The fastest growth in the population in dacha-recreational villages and new residential areas occurred in the 2010–2019 decade (Figure 4).

Considering spatial features and dynamics, it is possible to conclude that the entire metropolitan area experienced a relatively rapid population de-concentration after the dissolution of the USSR. Whereas the urban core accounted for 76.8% of the total population of the metropolitan area in 1989, it had dropped to 69.2% by 2021. In-migration was the main driver of population change. Overall, the most attractive suburban municipalities for migrants and the fastest-growing population appear to be those located north and east of the urban core in the Vsevolozhsk district of the Leningrad region. Only 16 out of 111 municipalities in the suburbs of Saint Petersburg metropolitan area have experienced significant population outflow in recent years, including both municipalities located on the periphery and with poor transport accessibility to the urban core and municipalities that have an advantageous geographical position. The latter may become areas of mass housing construction and witness significant in-migration shortly. Considering functional types of suburban settlements, the old industrial suburbs and rural areas have seen relatively less migration growth than the urban core. Population increase in dacha-recreational villages was 5–7 times higher than in the core. In the new residential areas, only according to official

(underestimated) data, the population, due to migration, has increased annually by more than 10%. Given that, the rate of total population growth in the municipalities of this functional zone over the past ten years has been, on average, higher than 20% per year.

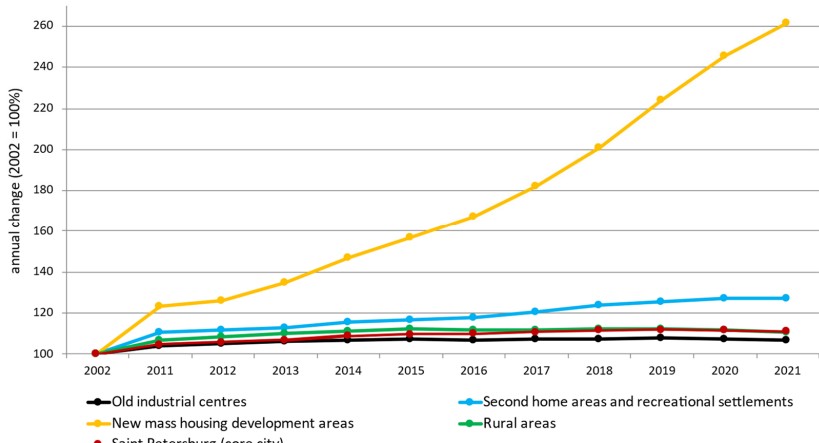

**Figure 4.** Dynamics of population change in the Saint Petersburg metropolitan area by settlement type.

**Riga metropolitan area.** During the observed decades, the Riga metropolitan area was characterized by significant population growth as compared with other non-metropolitan regions of the country, where the population generally declined. The most evident increase can been observed in the municipalities close to the capital city, wherein the population has even doubled in some areas. In the periphery of the territory, suburbanization processes are less pronounced, and there is population decline. The urban core of the metropolitan area, i.e., the cities of Riga and Jurmala, also witnessed population decline. Population changes in the Riga metropolitan area have been mainly driven by migration. The suburban municipalities are a national migration hub, attracting the residents of Riga as well as those moving closer to the capital from other regions of the country.

Comparing the studied settlement types, there has been a steady increase in population in the rural areas of suburbs but no significant change in population in the suburban cities and towns (Figure 5). The highest in-migration and population increases are observed in the proximity of Riga and in purely rural areas. Rural areas of the suburbs are characterized by a significantly younger population than Riga city, a trend that has become slightly more pronounced over the past 20 years. Suburban cities and towns have a similar age structure to Riga, with a higher proportion of elderly residents as compared with rural parts of the metropolitan area.

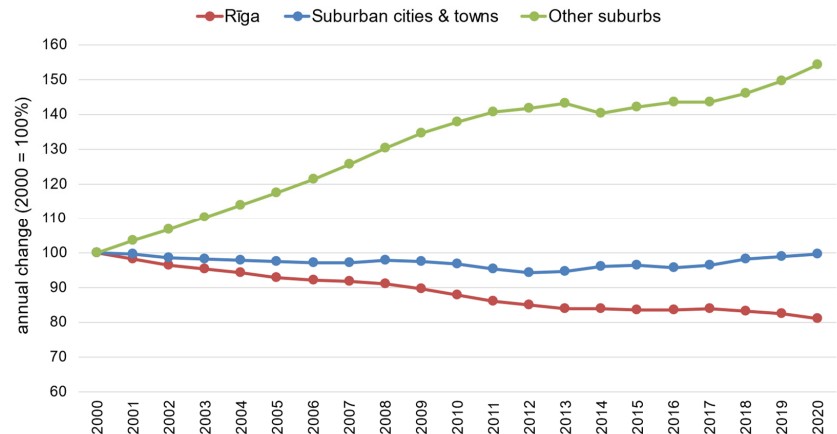

**Figure 5.** Dynamics of population change in the Riga metropolitan area by settlement type.

*5.2. Socioeconomic Status*

Socioeconomic wealth has been measured by variations in average income figures in the Latvian case. Income is one of the most used proxies for societal change in residential settings [21]. Municipal average income data of residents are not available for Russia. Municipal revenues are widely used in the urban economics literature as proxies for average income [77–79], but their application to the Russian context requires caution due to the significant role of transfers and subsidies from the national government budget; therefore, we have adopted only housing tax, i.e., a local revenue based on the value of property, as approximate measure of wealth; however, data for this measure were available only for the year 2016 (Figure 6).

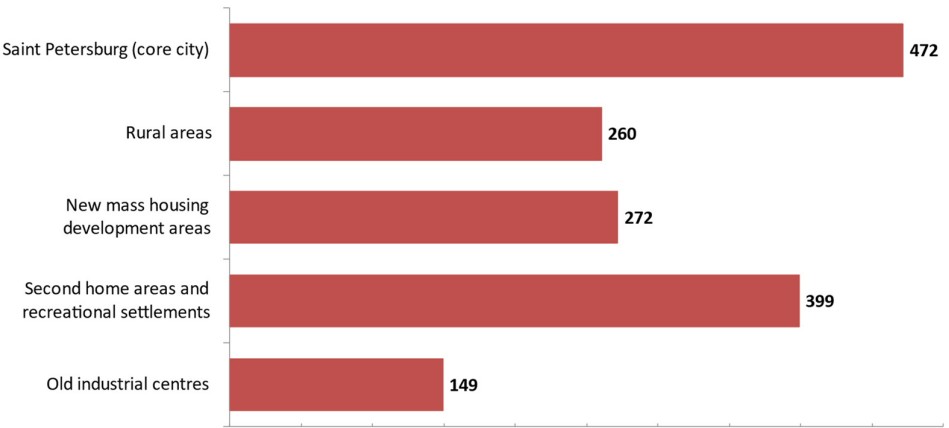

**Figure 6.** Average municipal housing tax (2016) by settlement type in the Saint Petersburg metropolitan area.

**Saint Petersburg metropolitan area.** We considered the amount of real estate tax paid by homeowners (rubles per resident) to be a measure of wealth. Data about real estate taxes paid to the municipal budget allow to estimate the cadastral value of housing, which is the main asset in Russia for most citizens. Most of the suburban municipalities with a high property tax value are territories located to the north of the urban core on the Karelian Isthmus, an environmentally attractive and traditional prestige resort area characterized by expensive dacha-style country houses, where summer vacationers from Saint Petersburg predominate among property owners. Within the boundaries of the suburban area, the lowest property tax is levied in rural areas, and the highest in areas of new single housing development. This allows us to speak of micro-level social polarization within rural areas, where wealthier and poorer residents are neighbors.

**Riga metropolitan area.** Over the past decade, the average income of the population, measured by monthly salary, has increased significantly across the metropolitan area. The Riga metropolitan area is generally more prosperous than other non-metropolitan regions of the country. Looking at the analyzed areas or settlement types separately, there has been a steady increase in average monthly salary over the past decade, but no major differences between the types. For example, suburbanized rural areas have experienced only slightly higher income growth than Riga and suburban towns. Even though a gradient factor could be identified in 2011, the gap between wealthier and poorer municipalities does not appear to have grown as a result of suburbanization; in some areas of intense development, the increase in average income is actually smaller than the average. These findings could point to a saturation of development potential or attractiveness, and they could be linked to recent re-urbanization trends among middle-upper-class suburbanites.

*5.3. Housing Supply and Market Trends*

Housing supply and market trends were analyzed by examining the growth in construction of new residential buildings and recent housing prices over the decades covered. Housing prices were estimated in the Latvian case by considering the last decade trend

in terms of sales and prices per square meter and the most recent data about average transaction prices, whereas recent data from real estate agencies about prices per square meter were used in the case of Saint Petersburg.

**Saint Petersburg metropolitan area.** The dynamics of the construction of multi-apartment residential buildings over the past two decades, in general, follow the spatial features of population change (Figure 7). The highest rates of housing construction are observed in new mass housing areas (more than 4.1 times over the 2000–2019 period). During the same period, the total housing stock area in the old industrial centers of the suburban area increased by 17.2%, and in rural areas only by 9.0% as compared with the 1999 level. In dacha-recreational cities territories, the growth was +41.5%, and in the urban core of Saint Petersburg it was +57.3%. It draws attention to the fact that in all types of suburban functional areas, as well as in the urban core, the growth rate of the housing stock of apartment buildings significantly exceeded the growth rate of the population. This indicates an increase in the area of housing per capita in new buildings in recent years as compared with the Soviet period. In general, over the past twenty years, the housing stock of suburban municipalities has increased by 45%.

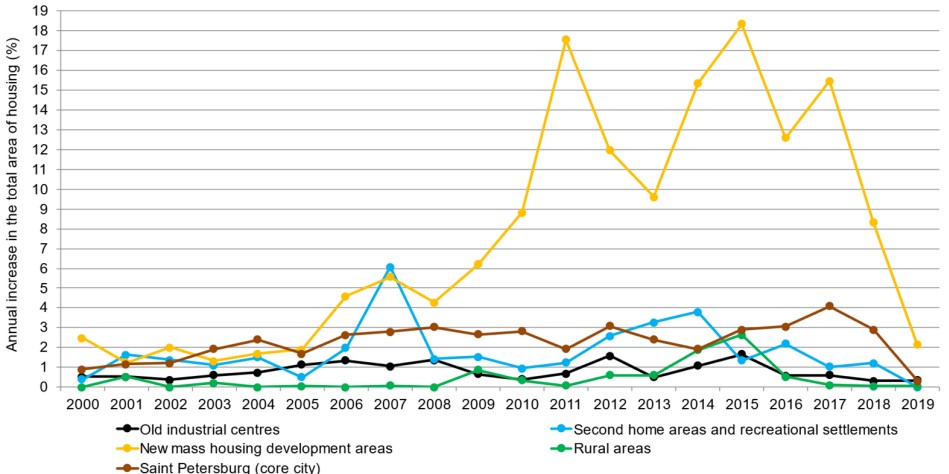

**Figure 7.** Housing construction dynamics by settlement type in the Saint Petersburg metropolitan area.

Differentiation trends are evident looking at one of the administrative districts of the Leningrad Region, i.e., the Vsevolozhsk District, which is adjacent to the urban core of Saint Petersburg and fully included in the suburban area. Over the past 20 years, the district territory has been subject to intensive residential development targeted at various socioeconomic groups. This process has been especially visible since 2010 in the municipalities which border the urban core of St. Petersburg. The analysis of the yearly construction share of apartment buildings shows that the share of housing built in the period 2010–2018 accounted for more than 50% of the total housing stock in some of these municipalities, which have consequently been 'promoted' from rural to urban settlement status during the decade. Conversely, in the most distant municipalities from the core the increase in the last decade has been less than 10%.

In order to investigate housing market trends, the average prices of housing sales through real estate agencies in the period from October 2019 to March 2020 were examined. As expected, the highest costs were found in the most prestigious suburban cities and towns of the metropolitan area located on the coast of the Gulf of Finland in the Kurortny (resort) district of Saint Petersburg federal city, as well as in a municipality including a famous tsarist palace and park complex, the city of Pushkin. With increasing distance from the urban core, costs gradually decrease from 1000–1100 euros per m$^2$ in core-bordering municipalities to 300–400 euros per m$^2$ in municipalities located 60–80 km from the core. The exceptions to this center-periphery tendency are relatively large cities of the Leningrad

Region in which real estate prices are higher than in small towns and rural settlements closer to the core.

There is a significant differentiation in the cost of housing among different settlement types, with an almost two-fold gap between the cost of 1 m$^2$ in rural areas and in areas of new single housing development (Table 1). Intra-municipal differentiation is also relevant. The analysis of housing prices in the Vsevolozhsk district shows that, whereas municipalities with the most intensive housing construction are distinguished by lower average prices with non-significant variations, in the areas of less intensive construction the cost of housing varies in a very wide range.

**Table 1.** Housing market prices in the Saint Petersburg metropolitan area.

| Settlement Type | Housing Prices (Eur) per 1 m$^2$ (2019–2020) |
| --- | --- |
| Saint Petersburg urban core | 1400 |
| Old industrial centers | 872 |
| Second home areas and recreational settlements | 1144 |
| New mass housing development areas | 1000 |
| Rural areas | 615 |

**Riga metropolitan area.** In the case of Latvia, the Riga metropolitan area housing market has seen a significant increase in demand for new housing, most markedly outside the urban areas. This is mainly confirmed by census data on the period of housing construction. Overall, the 2001–2007 construction boom had a significant impact on suburban residential development. However, the highest construction intensity can be observed in the rural areas of the Riga metropolitan area. It should be emphasized that the housing market and construction activity in the Riga metropolitan area is quite polarized, with Riga on one side and rural suburban areas on the other side.

The sharpest housing supply increase in the decade can be observed in the (mostly rural) municipalities in the immediate surroundings of Riga. Differently from Riga, Jurmala and traditional suburban towns, where the transaction volume has been basically stable since 2012–2013 (or decreasing, in the case of Jurmala), indeed, real estate transactions have increased in rural parishes of the functional area in the post-crisis years (Figure 8). Another characteristic that sets rural areas in the Riga metropolitan area aside from urban areas is the fact that land sales constitute an absolute majority of transactions, a trend that increased in the 2014–2020 period, whereas apartment sales still predominate in suburban towns.

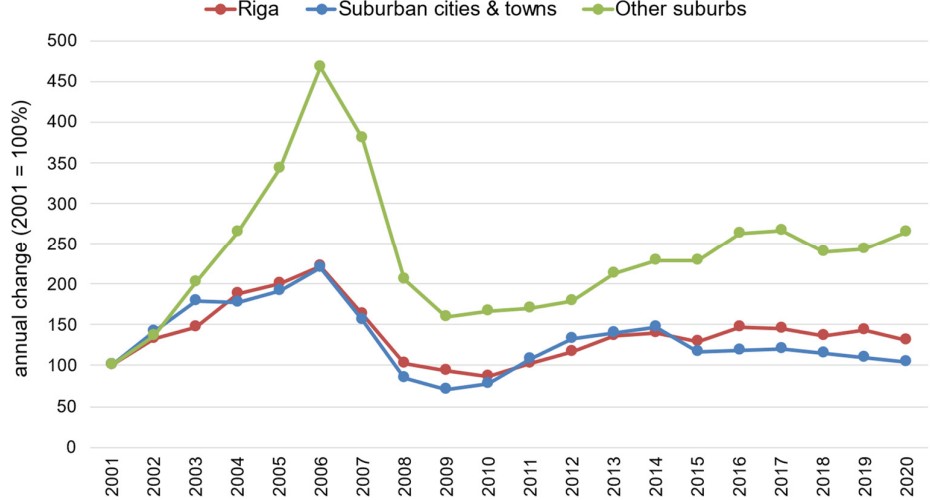

**Figure 8.** Dynamics in real estate sales by settlement type in the Riga metropolitan area.

Average prices for real estate transactions during the past decade show an overall price increase in suburban towns (despite an overall decrease in the volume of transactions) as well as rural parishes (Figure 9). The only evident decrease in prices is found in Jurmala, where the market has been affected by changes in the regulation for granting temporary residence permits to third-country nationals. The most expensive prices as of 2019–2020, in addition to those of Riga and Jurmala, are observed in the traditionally rural and suburban upper-class municipalities of Babite, Marupe, and Garkalne; in general, prices tend to decrease with the distance from Riga. (Table 2).

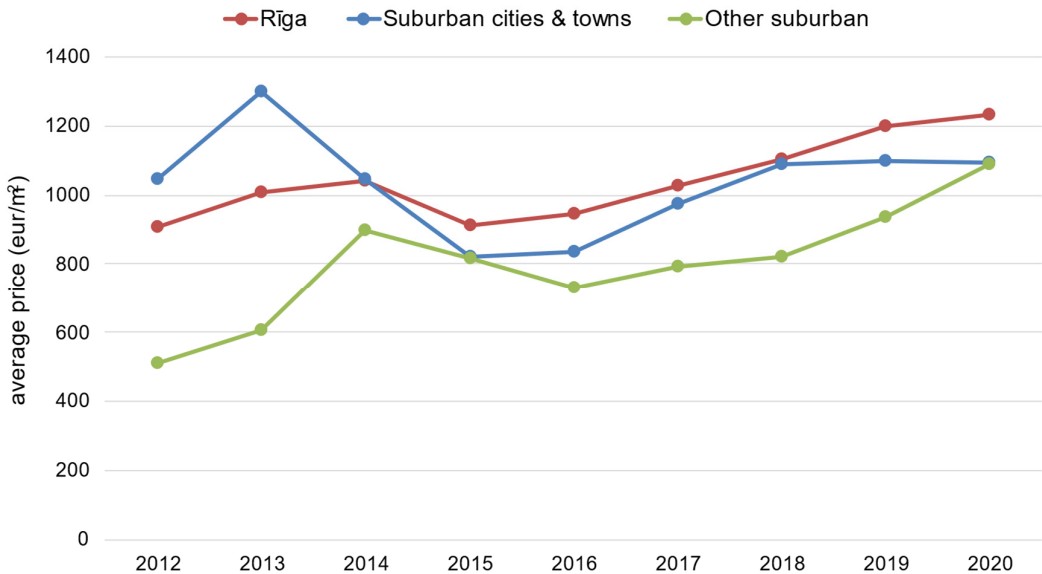

**Figure 9.** Dynamics of average housing prices by settlement type in the Riga metropolitan area.

**Table 2.** Housing prices in Riga metropolitan area.

| Settlement Type | Housing Prices (Eur) per Transaction (2020) |
|---|---|
| Riga | 62,600 |
| Jurmala | 116,100 |
| Suburban towns | 42,800 |
| Rural areas | 43,500 |

## 6. Discussion

The results of our study have evidenced that the suburbanization of St. Petersburg and Riga in the 2010s, as in the previous decade, has been a demographically and economically significant phenomenon, whose dynamics in the two cases present significant differences and analogies. Differences mostly relate to migration dynamics and to the short- and medium-term impact of the 2008 financial crisis on the housing market and construction industry.

Regarding migration dynamics, Saint Petersburg, differently from Riga, is characterized by significant in-migration, both internal and international. This explains, to a significant extent, the emergence of new mass housing development projects, targeted at lower-middle-class residents but also, and perhaps most relevantly, at in-migrants. As for the latter, the much faster pace of the housing market recovery in Russia explains the different intensities of housing supply development in the two suburban areas and reveals 'cyclic' patterns of investment and disinvestment in Latvia. These differences also explain why the uneven character of the phenomenon is evident to a different extent in the two cases.

In Saint Petersburg, the observed socio-spatial polarization relates to suburbanization in two ways: the differentiation of the housing market for different categories of suburban-

ites and the upgrading, which may be regarded as a form of suburban gentrification, of many traditional rural municipalities. Hence, the bifurcation between middle-upper-class dacha districts and lower-middle-class high-rise districts creates inter-municipal polarization, whereas the emergence of new expensive dacha developments in traditional rural, lower-class settlements, generate micro-level differentiation dynamics. Polarization trends, in particular inter-municipal trends, are less evident in the Latvian case, mostly due to the absolute prevalence of low-density suburbanization; however, these forms of development are also likely to contribute, to an extent, to intra- and inter-municipal polarization. On the one hand, new villa districts around Riga, which are developed in already existing rural/semi-rural settlements as well as in entirely undeveloped areas, are targeted at relatively wealthy suburbanites; on the other hand, limited development in suburban towns basically fulfils, on a much smaller scale, the same function as new mass housing developments around Saint Petersburg, as a 'budget' option for lower-middle-class residents and internal migrants. Finally, the decrease in income growth pacing in the Riga suburban area may be partially explained by the 'cyclic' disinvestment and re-investment factor—an incipient trend of re-urbanization of the middle-upper-class districts following increasing re-investment and gentrification dynamics in the inner-city core in the last years, after partial real estate market recovery.

Despite these relevant differences, analogies are evident and point to a co-presence of supply side and demand side dynamics when trends are looked at in the context of broader urban change. In both cases, the willingness of suburban municipalities to attract new residents/taxpayers makes suburban development constraints weaker than in the core city. In Saint Petersburg, constraints, investment risks, and lack of short-term profitability for large scale regeneration/gentrification in the historical core and Soviet mass housing maintenance and renovation, suburbanization, which are both attractive for the middle-upper-class residents and viable for lower-middle-class residents and internal migrants, clearly appears to be a suitable, low-risk option for investors and developers. In Riga, next to similar constraint factors to city core development, at least with regard to mass housing districts, the financial crisis also played a major role: The contraction of credit led to investments being directed away from the inner core of the capital towards rural suburban areas, with less expensive transactions and risks for the demand side and the supply side. However, partial recovery in the last years may imply a re-investment trend in the inner core.

From the point of view of the demand side, a mix of economic and societal factors need to be considered. On the one hand, budget suburban housing is certainly mostly a trade-off choice based on financial resources and constraints for immigrants (in the Saint Petersburg case) and lower-middle-class suburbanites, although distance, connection, and availability of services all play roles in defining 'attractiveness' and orienting choices. On the other hand, a societal aspiration component, perhaps more status-centric in the Russian case, can be identified in demand for detached housing; upcoming Census data may allow a clearer understanding of this dynamic and may provide a clearer understanding of the extent of suburbanization-related socio-spatial segmentation.

The dynamics of housing development in both metropolitan areas, heavily affected by the fast emergence of land commodification and profit-oriented development following the collapse of state socialism, also raise serious questions about the compatibility between development logic and sustainability goals. In addition to the effects of urban sprawl in terms of air pollution and transport challenges, the emphasis on suburban (and in-fill) development at the expense of regeneration of urban residential infrastructure significantly increases land consumption.

## 7. Conclusions

Our study aimed to investigate suburbanization trends and drivers in the metropolitan areas of Saint Petersburg and Riga, taking into consideration the broader processes of metropolitan changes in the two cases. Our pilot analysis corroborates the interpretation

of the significant suburban trend as the combination and interplay of production side dynamics and consumption side dynamics, with a significant role of demographic and housing market differences. Overall, bifurcation/polarization can be identified in both cases between detached housing areas and areas of mass housing development. The former are characterized by higher wealth indicators; the latter are characterized by higher construction and population increase rates around Saint Petersburg, but not around Riga, where the influx is much more modest and concentrated in pre-existing suburban towns. Regarding detached housing areas, in both cases, the influx of wealthy suburbanites can also foster micro-polarization dynamics.

On the one hand, the supply side search for locations that present fewer obstacles/risks to marketization and neoliberal dislocation is an evident phenomenon, which strongly underlines commonalities in the logic of urban-metropolitan change in the two cities. In Saint Petersburg, suburban development appears to be the most convenient supply side strategy, since inner city regeneration and renovation of mass housing present high investment risks or are seen as unprofitable. In Riga, in addition to this, supply side dynamics have been influenced by the contraction of the real estate market and the private credit system. The recent trend towards gentrification-driven re-urbanization of the inner-city core points out, once more, the role played by developers in cyclically influencing socio-spatial trends according to profit-focused investment logic.

On the other hand, demand side dynamics must be understood in terms of microeconomic strategies, i.e., trade-off considerations between cost and location due to the relative affordability of suburban housing and societal aspirations towards suburban life, mostly in the form of detached housing. Around Saint Petersburg, these elements are embodied by the segmentation of the housing market between new mass housing development and villa/cottage (dacha-recreational) districts, whereas, around Riga, the needs of the lower-middle-class segment are addressed by development in urban centers.

The characteristics of suburbanization and metropolitan change in the two cities have another important implication. Rather than showing a substantial diversity of segmentation dynamics in the cities of the former USSR and Eastern bloc as compared with the Western contexts which are predominantly analyzed in the literature, they emphasize the relevance of contextual adaptation of neoliberal transformations. In this regard, both hybridization and de-territorialization appear to be useful analytical frameworks. Hybridization emphasizes the globalized characteristics of post-socialist urban transformations; de-territorialization points out the very specific interplay of neoliberal forces and socialist era-inherited morphological and infrastructural characteristics. Therefore, our conceptual and methodological approach could be useful to unpack the forces and logic behind suburban development in other parts of the world, provided that contextual legacies and other specificities are taken into account.

It must be stressed that the development trajectories of Saint Petersburg and Riga cannot be considered to be universally representative of metropolitan and suburban development in the former USSR. The two cities have similar morphological characteristics (a concentric structure, typical of European cities with a pre-socialist historical core) and central roles in their respective national spatial development and economic systems. Planned cities and more peripheral urban centers are certainly affected by different contextual conditions; further comparative analysis would benefit from analyzing different urban and metropolitan typologies. Moreover, the central position of the two cities does imply their demographic dynamics are distinct from the shrinking patterns of peripheral urban areas in the former USSR and beyond, which face criticalities and opportunities of a different kind [80,81]. Nevertheless, our study underlines patterns and drivers of development which reflect a widespread logic and dynamics in the former Soviet space, and therefore, are likely to be significant for most post-Soviet metropolitan areas.

Our comparative study is also affected by data gaps and harmonization issues which, to an extent, affect the quality of results. This limit should necessarily be addressed by further research on the topic. The upcoming availability of data from the last Census

surveys, carried out in Latvia and Russia in 2021, are likely to provide very useful information in this regard, in particular with reference to the socioeconomic dimension of the suburbanization phenomenon.

Another timing issue is worth mentioning in terms of implications for future research. The outreach of the COVID-19 pandemic will also need to be taken into account as a relevant factor affecting short- and long-term suburbanization trends. The pandemic's social and economic impacts may provide a push towards further suburbanization that would counter, or possibly even overcome, centripetal trends.

**Author Contributions:** Conceptualization, G.S. and M.B.; methodology, G.S., Z.K. and D.Z.; formal analysis, M.B. and D.Z.; writing original—draft preparation, G.S. and D.Z.; writing—review and editing, G.S., D.Z. and M.B.; visualization, D.Z. and M.B.; supervision and project administration, Z.K. All authors have read and agreed to the published version of the manuscript.

**Funding:** This research was funded by the Latvian Council of Science, project DemoMig, project no. VPP-LETONIKA-2021/4-0002.

**Institutional Review Board Statement:** Not applicable.

**Informed Consent Statement:** Not applicable.

**Data Availability Statement:** Not applicable.

**Acknowledgments:** We are grateful to Alla Makhrova for her insightful comments on an earlier version of the manuscript.

**Conflicts of Interest:** The authors declare no conflict of interest. The funders had no role in the design of the study; in the collection, analyses, or interpretation of data; in the writing of the manuscript, or in the decision to publish the results.

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
