# Peer review of "Post-Soviet Suburbanization as Part of Broader Metropolitan Change: A Comparative Analysis of Saint Petersburg and Riga"

_sustainability, doi:10.3390/su14138201_

Round 1

Reviewer 1 Report

The paper deals with the phenomenon of suburbanization of the two post-socialist cities of St. Petersburg and Riga in the period after the world economic crisis in 2008.

This period is chosen having the specific demographic and economic changes that marked it and conditioned the development of certain patterns of suburbanization in the given cities.

Differences and analogies have been noticed under the influence of migrations and the impact of the economic crisis on the real estate market and the construction industry.

The work is important because it contributes to the understanding of suburbanization trends and post-socialist transformations within the broader processes of metropolitan changes, hybridization and deterritorialization. 

I hope the suggestions would help you.

best regards

Author Response

Point 1: Moderate English changes required

Response 1: we have carried out a thorough proof-reading of the text.

Further interventions

Following the Editor’s suggestions:

  1. we have inserted a paragraph about the context of population decline in CEE and the former USSR (lines 329-340) and recalled the topic in the Conclusions (lines 848-861);
  2. we have split Table 1 into two and reformulated the captions;
  3. we have inserted references to sustainability-related issues, in particular land consumption, in Sections 1 (lines 67-72) and 2 (lines 191-192), Study context (lines 378-382), and Discussion (lines 796-802).

Moreover, next to reworking Table 1, we have carried out a revision of Section 5 to make it more readable. In particular:

-we have revised information and graphs in sub-section 5.2

- we have added Figure 11 in order to illustrate the final text paragraph in sub-section 5.3

Reviewer 2 Report

The paper "Post-Soviet Suburbanization as Part of Broader Metropolitan Change: a Comparative Analysis of Saint Petersburg and Riga" is a rather interesting and original attempt to advance the knowledge on post-socialist urban change and suburban development. However, I find some important shortcomings that prevent the current version of the article from being accepted for publishing.

This manuscript hardly fits the scope of the Special Issue (as well as Sustainability in general). In the study, the reader would expect at least some discussion related to the phenomenon of sustainability (regardless of how the authors interpret it). I would recommend to (re)orient the aim of the study in such a way that it better meets the purpose of the special issue.

The study sets a very ambitious aim - to perform multi-scalar, multi-dimensional and comparative study, however data limitations do not allow it to be achieved. Data problems are highlighted in the paper, but it is still one of the main obstacles to achieve scientific soundness and significance as well as overall merit. Alternatively, it might be possible to limit the aim to something like - to set the theoretical-conceptual foundations and complete the pilot study on Riga and Saint Petersburg).

While this is a unique exercise to compare the two cities (that are very different!), the argumentation is missing why these two cities are compared. What does (could) it say about other (post-soviet) cities?

The manuscript is about post-socialist urban change and it is therefore very regional in geographical terms; it contains very little hint as to why the knowledge gained may be relevant to the rest of the world; for the broader audience. The article is also very descriptive, part of the results is a report of the statistical data.

The presentation of data in the figures requires a lot of adjustments and additional efforts to make them comparable and easier to understand (given the current data challenges). 

Author Response

Point 1: This manuscript hardly fits the scope of the Special Issue (as well as Sustainability in general). In the study, the reader would expect at least some discussion related to the phenomenon of sustainability (regardless of how the authors interpret it). I would recommend to (re)orient the aim of the study in such a way that it better meets the purpose of the special issue.

Response 1: we have added references to the connection between suburbanization and sustainability issues (land consumption in particular) in the Introduction (lines 67-72), Section 2 (191-192), Study context (lines 378-382), and Discussion (lines 796-802).

Point 2: The study sets a very ambitious aim - to perform multi-scalar, multi-dimensional and comparative study, however data limitations do not allow it to be achieved. Data problems are highlighted in the paper, but it is still one of the main obstacles to achieve scientific soundness and significance as well as overall merit. Alternatively, it might be possible to limit the aim to something like - to set the theoretical-conceptual foundations and complete the pilot study on Riga and Saint Petersburg).

Response 2: we have added references to the pilot nature of our empirical analysis in the Abstract (line 18), Introduction (lines 144/154-156), and Conclusions (line 806).

Point 3: While this is a unique exercise to compare the two cities (that are very different!), the argumentation is missing why these two cities are compared. What does (could) it say about other (post-soviet) cities?

Response 3: we have added a paragraph in the Conclusions (lines 845-858) about both the usefulness and limits of our case studies in terms of extending our findings to the broader post-Soviet urban and metropolitan context.

Point 4: The manuscript is about post-socialist urban change and it is therefore very regional in geographical terms; it contains very little hint as to why the knowledge gained may be relevant to the rest of the world; for the broader audience. The article is also very descriptive, part of the results is a report of the statistical data.

Response 4: we have added explicit reference to the significance of the study for the global context, connected to the concept of deterritorialization, in the Conclusions (lines 841-844).

Point 5: The presentation of data in the figures requires a lot of adjustments and additional efforts to make them comparable and easier to understand (given the current data challenges).

Response 5: we have harmonized the maps in Figures 1 and 2

Further interventions

Following the Editor’s suggestions:

  1. we have inserted a paragraph about the context of population decline in CEE and the former USSR (lines 329-340) and recalled the topic in the Conclusions (lines 848-861);
  2. we have split Table 1 into two and reformulated the captions.

Moreover, next to reworking Table 1, we have carried out a revision of Section 5 to make it more readable. In particular:

-we have revised information and graphs in sub-section 5.2

- we have added Figure 11 in order to illustrate the final text paragraph in sub-section 5.3

Reviewer 3 Report

The discourse and logic of the Introduction are clear, referring mainly to the global debate over the types of transformations that have taken place in post-communist and in particular post-Soviet urban areas. However, I believe that the authors could also draw inspiration from similar studies aimed at urban transformations in post-communist states outside the former Soviet space. Here I am referring to methodological aspects or types of spatial transformations, which in turn involve suburbanization, urban sprawl, etc. There are many illustrative examples in this regard in countries such as the Czech Republic, Hungary, Romania, Poland, etc. I think it will be able to help authors expand (at least conceptually) the type, dynamics and patterns of change. These are summarized in the section dedicated to the sub-urbanization debate, but rather in a conceptual manner, without providing illustrative examples. However, the section dedicated to the sub-urbanization debate is very interesting and useful.

The study context presents clearly and explicitly the political, economic and social context in which the changes related to land use and spatial planning took place.

There is some repeatability between 3. The Study Context and 4.1. The Case Study Approach + 4.1. The Study Area: Spatial Extent of the Selected Metropolitan Areas. I kindly ask the authors to try to fix these aspects.

Please try to have a consistency between the maps in Figures 1 and 2 in terms of including common elements, having a similar legend, using similar colours. Also, try to find a common ground for the rest of the comparative figures, e.g., Figures 4 and 5, Figures 6 and 7.

Please translate the legend of Figure 9 into English.

I would suggest the authors to separate Discussions from the Conclusions to better highlight the significance of the findings in light of what was already known in the literature about the research problem (in the case of Discussions). Also, the conclusions should highlight the main outcomes of the paper in terms of some quantitative values.

Author Response

Point 1: The discourse and logic of the Introduction are clear, referring mainly to the global debate over the types of transformations that have taken place in post-communist and in particular post-Soviet urban areas. However, I believe that the authors could also draw inspiration from similar studies aimed at urban transformations in post-communist states outside the former Soviet space. Here I am referring to methodological aspects or types of spatial transformations, which in turn involve suburbanization, urban sprawl, etc. There are many illustrative examples in this regard in countries such as the Czech Republic, Hungary, Romania, Poland, etc. I think it will be able to help authors expand (at least conceptually) the type, dynamics and patterns of change. These are summarized in the section dedicated to the sub-urbanization debate, but rather in a conceptual manner, without providing illustrative examples. However, the section dedicated to the sub-urbanization debate is very interesting and useful.

Response 1: we have inserted references to studies about the socio-spatial and sustainability implications of suburbanization in cities of central-eastern Europe in Section 2 (lines 177-191).

Point 2: There is some repeatability between 3. The Study Context and 4.1. The Case Study Approach + 4.1. The Study Area: Spatial Extent of the Selected Metropolitan Areas. I kindly ask the authors to try to fix these aspects.

Response 2: we have renamed sections 4.1 and 4.2 respectively as Methodology and Spatial Extent of the Selected Metropolitan Areas.

Point 3: Please try to have a consistency between the maps in Figures 1 and 2 in terms of including common elements, having a similar legend, using similar colours. Also, try to find a common ground for the rest of the comparative figures, e.g., Figures 4 and 5, Figures 6 and 7.

Response 3: we have harmonized the maps in Figures 1 and 2.

Point 4: Please translate the legend of Figure 9 into English.

Response 4: we have translated the legend into English.

Point 5: I would suggest the authors to separate Discussions from the Conclusions to better highlight the significance of the findings in light of what was already known in the literature about the research problem (in the case of Discussions). Also, the conclusions should highlight the main outcomes of the paper in terms of some quantitative values.

Response 5: we have separated the Discussion and the Conclusions. In the Conclusions, we have added a paragraph stressing the demographic and socio-economic polarization aspects of suburbanization in the two metropolitan areas according to quantitative data (lines 811-817).

Further interventions

Following the Editor’s suggestions:

  1. we have inserted a paragraph about the context of population decline in CEE and the former USSR (lines 329-340) and recalled the topic in the Conclusions (lines 848-861);
  2. we have split Table 1 into two and reformulated the captions;
  3. we have inserted references to sustainability-related issues, in particular land consumption, in Sections 1 (lines 67-72) and 2 (lines 191-192), Study context (lines 378-382), and Discussion (lines 796-802).

Moreover, next to reworking Table 1, we have carried out a revision of Section 5 to make it more readable. In particular:

-we have revised information and graphs in sub-section 5.2

- we have added Figure 11 in order to illustrate the final text paragraph in sub-section 5.3

Round 2

Reviewer 2 Report

Dear author(s), thank you for responding to earlier comments and for providing a very clear overview of your responses. Overall, I think this helped to improve the paper. I have some following recommendations:

1) Although, there is no restrictions on the length of the manuscript, the text could be more concise. 

2) I still have doubts about the figures; their comparability and design could be improved further. E.g., Figure 3 could use the same type of legends and (exactly) the same color scheme. The utility of Figure 7 is questionable - is it really worth to include it? In the legend of Figure 1 and 8 "Saint Peterburg" is misleading - I assume it is the "core city"? Similarly in Figure 2: "Capital city - Riga" - does it indicate administrative borders or core city?  

Author Response

Point 1: Although, there is no restrictions on the length of the manuscript, the text could be more concise. 

Response 1: We have carried out further editing of the text and eliminated some redundancies.

Point 2: I still have doubts about the figures; their comparability and design could be improved further. E.g., Figure 3 could use the same type of legends and (exactly) the same colour scheme. The utility of Figure 7 is questionable - is it really worth to include it? In the legend of Figure 1 and 8 "Saint Peterburg" is misleading - I assume it is the "core city"? Similarly in Figure 2: "Capital city - Riga" - does it indicate administrative borders or core city?  

Response 2: We have harmonized colour schemes and types of legends of both maps in Figure 3. In assessing the utility of Figure 7, we agree that this figure is not very useful in interpreting the data. In Figure 1 and 8, St.Petersburg has been replaced with ‘Saint Petersburg (core city)’. This designation is also specified in all the other relevant figures. In Figure 2, Riga borders represent administrative borders.
